# Nightmares’ Pattern and Predictors among a Saudi University Students during COVID-19 Pandemic

**DOI:** 10.3390/ijerph19031776

**Published:** 2022-02-04

**Authors:** Amal A. Alghamdi, Nora O. Alafif, Ahmed S. BaHammam, May Almuammar, Najd S. Alharbi, Nouf A. Alhefdhi, Sara Al-Musharaf, Kholoud S. Al Ghamdi, Ghada F. AlYousif, Norah F. Al Muhanna

**Affiliations:** 1Department of Family and Community Medicine, College of Medicine, Imam Abdulrahman bin Faisal University, Dammam 34212, Saudi Arabia; amlalghamdi@iau.edu.sa (A.A.A.); gfalyousif@iau.edu.sa (G.F.A.); 2Department of Community Health Sciences, College of Applied Medical Sciences, King Saud University, Riyadh 10219, Saudi Arabia; malmuammar@ksu.edu.sa (M.A.); alharbinajd978@gmail.com (N.S.A.); nouf.alhefdhi@gmail.com (N.A.A.); salmosharruf@ksu.edu.sa (S.A.-M.); 3 The University Sleep Disorders Center, Department of Medicine, College of Medicine, King Saud University, Riyadh 11324, Saudi Arabia; ashammam2@gmail.com; 4The Strategic Technologies Program of the National Plan for Sciences and Technology and Innovation in the Kingdom of Saudi Arabia, Riyadh 11362, Saudi Arabia; 5Department of Physiology, College of Medicine, Imam Abdulrahman bin Faisal University, Dammam 34212, Saudi Arabia; ksaalghamdi@iau.edu.sa; 6Psychiatry Department, Imam Abdulrahman Bin Faisal University, Dammam 34212, Saudi Arabia; nfalmhana@iau.edu.sa

**Keywords:** nightmares, anxiety, depression, coronavirus, quarantine, university

## Abstract

Purpose: During the coronavirus disease 2019 (COVID-19) pandemic quarantine, university students were under various types of stressors, including the exams period, which might have affected their quality and quantity of sleep, and consequently, their quality of life. This study aimed to investigate the pattern and predictors of nightmares among university students and coinvestigate the presence of other types of sleep disturbances, mental disorders, and quarantine-related stressors. Methods: This cross-sectional study included 368 university students who answered a self-completed questionnaire covering their sociodemographic features, nightmare indicators, and associated quarantine stressors. Additionally, sleep disturbances were measured using the Generalized Sleep Disturbance Scale (GSDS), anxiety using the Generalized Anxiety Disorder 2 scale, and depression using the Patient Health Questionnaire-2. Results: The participants’ mean age was 20.4 ± 1.6 years, and male participants represented 35.9% of the sample. Nightmares were experienced by 117 (31.8%) of the participants, of whom 44.4% had new-onset nightmares. The mean GSDS was 45.0 ± 14.9 (min. = 12, max. = 130). This value is associated with elevated odds of the following outcomes: the presence of nightmares (odds ratio [OR] = 1.8; confidence interval [CI] 95% = 1.1–3.0); new-onset nightmares at the time of pandemic (OR = 2.6; CI 95% = 1.3–5.5); and anxiety (OR = 1.74; CI 95% = 1.0–2.9). The presence of nightmares elevated the score of GSDS by 11.3 points (S.E. = 1.6, *p* < 0.001), elevated the odds of anxiety by 4.1 (CI 95% = 2.5–6.8), and depression by 2.1 (CI 95% = 1.3–3.4). Conclusions: Stressors resulting from both the exams period and the fact that it was conducted during COVID-19 quarantine increased the rate and affected the pattern of nightmares. These stressors also led to other sleep disturbances and mental disorders that were significantly more prevalent among females.

## 1. Introduction

Saudi Arabia’s efforts to control the spread of the coronavirus disease 2019 (COVID-19) demonstrated the success of a chain of adopted measures and was described as a “success story” [1,2].

In early March 2020, Saudi Arabia announced its first confirmed COVID-19 case, after which the number of COVID-19 cases increased dramatically. By 23 March 2020, Saudi Arabia announced a quarantine plan. The quarantine was in effect for several months until it was gradually lifted, from 28 May 2020 onward. During the quarantine, Saudi residents were placed under various restrictions, including social and religious restrictions. One of the most vulnerable groups were university students, who were affected by various types of stressors, including their academic examination period, sociodemographic factors (i.e., income), social restrictions (i.e., restrictions on visiting family and friends), and religious practice restrictions (i.e., visiting mosques). The magnitude of the effect of these stressors on participants was assumed to be dependent on their role in the community. The more the person is responsible for financial support to the family, the more he/she is affected by the consequences of the pandemic. In addition, the media’s daily updates on COVID-19, reporting the dramatic spread of the virus and its related casualties, created a stressful situation that could be reflected in dreams [3]. Moreover, the quarantine period was also an examination period, and students were required to take their final examinations online for the first time, which is another factor that could increase their worries and be reflected in their sleep and dreams.

Dreaming is a complex cognitive state affected by various medical, psychological, sleep, and social variables [4]. Nightmare disorder is the most prevalent rapid eye movement (REM) parasomnia that involves a variety of dysphoric emotions, including fear in particular; nightmare disorder is one of the common symptoms of sleep disorders and is often associated with poor sleep quality [4]. Nightmares are defined as disturbing mental experiences that generally occur during REM sleep and often result in awakening [5]. The nightmare phenomenon is believed to increase during times of stress and has been linked with mental disorders [6,7]. During the COVID-19 pandemic, researchers reported that the pandemic news and associated quarantine restrictions might precipitate a stressful situation leading to several psychological disorders [8]. In addition, the literature described a reflection of the pandemic stressors on dream content and nightmare frequency. Recent data indicate that nightmares might be associated with the COVID-19 pandemic [9,10,11]. However, nightmares in vulnerable groups such as students have not been studied, and it is not clear if the characteristics of nightmares during these stressful times differ between males and females, as well as between females with or without pregnancy or hormonal supplementations [12]. Sex differences in relation to nightmares were associated with the frequency of nightmares [9,11]. However, the reasons why females suffered from more frequent nightmares than males were not totally clear. These differences could be attributed to the effect of underlying poor sleep quality, underlying psychological disorders, and/or previous psychological trauma.

In this regard, we hypothesized that the increased frequency of nightmares during the pandemic is associated with sleep disturbances and poor mental health; however, these nightmare-frequency-related changes affected males and females to varying extents during the pandemic quarantine. In addition, we suspected that the main theme of the participants’ nightmare could be associated with the COVID-19 pandemic (such as the loss of loved ones or contracting COVID-19) [9,11,13,14].

The effects of the COVID 19 pandemic on mental health, sleep, and nightmares are becoming of growing interest, especially after 2020, and there is an increasing number of studies tackling this matter. However, in KSA, there is limited literature that examines the effect of the quarantine measures that were taken by the Saudi government to contain the pandemic on the mental health, sleep, and nightmares of students. To the best of our knowledge, no study has examined the differences in the rate and pattern of nightmares between male and female students during the COVID-19 pandemic in relation to various predictors. Hence, this study sought to describe the pattern of nightmares during the COVID-19 pandemic among male and female university students and the related risk factors. We also aimed to investigate the relationship between nightmares, sleep disturbances, anxiety, and depression during quarantine and the impact of restriction-related pressures on study outcomes.

## 2. Methods

This cross-sectional study was conducted in Riyadh City, Saudi Arabia, after obtaining the required approval from the institutional review board of King Saud University (KSU) (Ref. No. 21/0025/IRB) using an online self-completed questionnaire distributed to students of KSU after obtaining their written informed consent. To reach students from all the KSU colleges, from different academic tracks (Sciences, Health, and Humanities), and decrease possible selection bias, the questionnaire was distributed randomly among KSU students by approaching KSU’s student representatives who contacted the leaders of each academic year and then shared an online link to a randomly selected sample (random number tables) of peer students’ academic tracks (Sciences, Health, Humanities). The data collection process lasted for two weeks at the end of May 2020, immediately before the quarantine started to be gradually lifted.

### 2.1. Sample Size Calculation

The sample size was calculated using Epi Info 7 software, analyzing 382 participants by applying the following information; the total number of bachelor degree students in 2020, 61,704; the allowed margin of error, 5%; the acceptable confidence level, 95%; the assumed rate of new-onset nightmares, 50% [15].

### 2.2. Questionnaire

The questionnaire consisted of 4 parts. The first part included questions regarding sociodemographic features and health indicators, including age; sex; marital status (married or single); parents’ level of education (primary education, secondary education, high school, university, and postgraduate); chronic diseases such as diabetes mellitus, hypertension, or respiratory diseases (yes or no), smoking (yes or no), and practicing regular exercise for 30 min 3 times per week (yes or no). The second part evaluated nightmares.

Nightmares were screened according to the Third Edition of the International Classification of Sleep Disorders (ICSD-3) diagnostic criteria, including recurrent episodes of awakening from sleep with a recall of dysphoric well-remembered dreams (with full alertness on awakening and good recall of sleep mentation). Upon awakening, the person rapidly becomes oriented and alert. Moreover, the disturbing dreams or the sleep disturbances produced by awakening result in clinically significant distress or impairment in social, occupational, or daytime function or delayed return to sleep after the episode.

The third part of the questionnaire assessed sleep disturbances among participants in relation to the COVID-19 pandemic using the validated Arabic version of the Generalized Sleep Disturbance Scale (GSDS), which consists of 21 items aimed to identify sleep disturbances in relation to the seven components of sleep. These components are: difficulty falling asleep (1 question), waking up during sleep (1 question), waking too early from sleep (1 question), quality of sleep (3 questions), the quantity of sleep (2 questions), fatigue and alertness at work (7 questions), and the use of substances to induce sleep (6 questions). Each component is scored from 0 to 7, with 7 indicating the highest frequency of sleep difficulties in the previous week. A frequency of at least 3 times per week was considered as significant sleep difficulties. The GSDS global score was computed by adding the scores of the seven sleep components [16].

The fourth component aimed to screen respondents for possible anxiety or depressed mood. Depressed mood was assessed using a validated Arabic version of the Patient Health Questionnaire-2, with 3 as the cutoff point for the presence of depression [17,18]. Anxiety was screened using a validated Arabic version of the Generalized Anxiety Disorder 2 scale [19], with a score of 3 or more indicating anxiety [20]. In addition, this part evaluated quarantine-related restrictions and their possible contribution to anxiety or nightmares. The participants were asked questions about the development of worries or concerns due to restrictions on their religious activities (such as visiting mosques), visiting their family members, and visiting their friends. The response to these questions was put on a Likert scale, with a score of 5 indicating maximum concerns in terms of restricting visits by family and friends and religious activities. Academic-related tension was evaluated by measuring the following academic indicators: current academic level, school (Sciences, Health, and Humanities), academic GPA (A, B, C, and D), and failing a subject (yes or no).

### 2.3. Statistical Analysis

The analysis was performed using STATA 16 (StataCorp, College Station, TX, USA). The chi-square (χ^2^) test, *t*-tests (*t*), and Fisher’s exact test were used to examine the differences in sociodemographic features and sleep variables between males and females, and logistic regression models were used to assess the risk of developing nightmares, anxiety, and depression in relation to several risk factors. Additionally, linear regression models were used to estimate the change in GSDS score in relation to various associations. The adjustment of various regression models was conducted using a directed acyclic graph, a hypothetical model representing the inter-relationship between the study variables. Each variable is represented by a node, and the relationship between the variables is represented by a unidirectional arrow [21]. A variable was included in the regression models as a possible confounder if it was graphically allocated between the exposure and the outcome and had a direct influencing relationship on both.

## 3. Results

A total of 368 participants were included in this study, corresponding to a response rate of 96%. The hypothetical directed acyclic graph used in the regression model adjustment is shown in Figure 1. The various regression models’ minimum sufficient adjustment (i.e., the minimum set of variables that were required to adjust for in order to illuminate confounding bias) included the following variables: academic level, sex, socioeconomic status, chronic diseases, smoking, and regular exercise.

Table 1 summarize the distribution of sociodemographic and health indicators among the male and female study participants. Most participants were females (*n* = 236, 64.13%), single (*n* = 356, 96.74%), and in their senior year (*n* = 256, 69.57%). The mean age of the participants was 20.4 ± 1.6 years (range: 18–26 years), 67 (28.4%) females reported irregular menses, and 8 (2.2%) females used oral contraceptives.

Table 2 summarize the distribution of nightmares across male and female groups. Most participants did not suffer from the presence of nightmares during the quarantine period (*n* = 251, 68.21%); however, (*n* = 52, 20.72%) experienced new-onset nightmares when the quarantine started. The presence of nightmares was more significant among females (*n* = 86; 73.8%) than males (*n* = 31; 26.5%, χ^2^(1) = 6.5; *p* = 0.010).). However, there were no statistically significant differences in the rate of nightmares between females who either experienced irregular menses or were taking oral contraceptives (*n* = 28; 32.6%) and other females (*n* = 58; 67.4%; χ^2^(1) = 0.9; *p* = 0.3). Regarding nightmare content, as seen in Table 2, the most common theme of nightmares during the pandemic was the “not related to COVID-19” nightmare (*n* = 147, 86.98%), as the majority of participants reported having “supernatural creatures or unexplained emotions” related nightmares (*n* = 50, 34.01%). However, most of those participants who had a “COVID-19 related nightmare” theme were females (*n* = 15, 68.18%). On the other hand, concerning male participants, the most common theme amongst the “COVID 19 related nightmare” was the “loss of loved one” theme (*n* = 7, 57.1%). Additionally, many participants with nightmares claimed that they were not bothered by having a nightmare (*n* = 76, 44.45%), although many of them reported staying awake sometime before going back to sleep (*n* = 79, 46.75%). When using the logistic regression analysis model, the odds of having nightmares or developing new-onset nightmares during the pandemic were not significantly increased with quarantine-related restrictions (i.e., restrictions on visiting families, friends, or mosques), sociodemographic features, and health indicators before or after adjusting for confounders (except for sex variable). The female sex increased the odds of experiencing nightmares (odds ratio [OR] = 1.8; 95% confidence interval [CI] = 1.1–3.0) and also increased the odds of developing new nightmares during the COVID-19 pandemic (OR = 2.6; 95% CI = 1.3–5.5).

Concerning sleep, the mean GSDS was 45.0 ± 14.9 (range: 12–130). There was no difference in the mean GSDS between males and females. On the contrary, there was a statistically significant difference in the GSDS score between females who were experiencing menstrual disturbances or using hormonal contraceptives (mean = 49.0 ± 4.7) and females without possible menstrual disturbances or usage of hormonal contraceptives (mean = 44.9 ± 14.0; *t*(234) = −2.0; *p* = 0.043). The mean subjective reported sleep duration (total daily sleep time) was 7.98 ± 1.8 h; (range 4–15 h). There was a statistically significant difference in the mean sleep duration between males (mean = 7.7 ± 1.6 h) and females (mean = 8.1 ± 1.8 h; *t*(366) = −2.2; *p* = 0.029). Approximately 33.4% (*n* = 123) of the participants reported taking naps during the day (mean = 1.7 ± 0.96 h/d; range: 0–5 h/d). Most participants reported poor sleep quality (*n* = 226, 61.41%). In addition, as summarized in Table 3, the majority of male participants with statistically significant nightmares reported frequent early waking from sleep (*n* = 24, 58.54%) and frequent daytime sleepiness (*n* = 20, 48.78%). On the other hand, female participants with statistically significant nightmares reported frequent difficulty falling asleep (*n* = 92, 71.87%), frequent waking up during sleep (*n* = 84, 65.62%), and frequent daytime sleepiness (*n* = 72, 56.25%). In addition, when using linear regression, it was found that the presence of nightmares increased the GSDS score by 1.29 points (S.E., 1.6; *p* < 0.001; adjusted R^2^ = 0.1). However, the onset of new nightmares in relation to the COVID-19 pandemic did not have a statistically significant influence on the GSDS score. Table 3 summarize the distribution of GSDS components among males and females and nightmares.

The anxiety screening results were positive in 27.2% (*n* = 100), and the depression screening results were positive in 34.5% (*n* = 127) of the participants. The anxiety rate was significantly higher among female participants (*n* = 73; 30.9%) than among male participants (*n* = 27; 20.4%; χ^2^(1) = 4.7; *p* = 0.030). There was no significant difference between males (*n* = 38; 28.8%) and females (*n* = 89; 37.7%; χ^2^(1) = 2.98; *p* = 0.084) in the distribution of depression. In addition, the anxiety rate among females with possible menstrual disturbances or hormonal contraceptives was 38.24% (*n* = 26), and the depression rate was 36.8% (*n* = 25). However, there were no statistically significant differences between these reported rates and the rates among females without menstrual disturbances or hormonal contraceptives. Table 4 and Table 5 summarize the relationship between developing anxiety, depression, and nightmares using logistic regression models.

Concerning academic pressure indicators, participants attending the sciences track showed the highest rate (significantly) of anxiety among students (*n* = 15; 50%; χ^2^(3) = 10.51; *p* = 0.015), whereas the students in the humanities track showed the lowest level of anxiety (*n* = 16; 19.3%). However, there was no statistically significant association between other academic indicators and depression and anxiety. Additionally, there was no statistically significant association between anxiety or depression and worries created by religious activities restrictions. However, there was a significant difference between males and females in the restrictions of visiting friends and family, as 44.88% (*n* = 57; χ^2^(3) = 8.2; *p* = 0.042) of participants with depression and 47.00% (*n* = 47; χ^2^(3) = 8.8; *p* = 0.032) of participants with anxiety were severely worried about not being able to visit their family or friends during the quarantine. Interestingly, this difference was significant among males but not females.

## 4. Discussion

This study examined the rate and pattern of nightmares among male and female university students during COVID-19 quarantine. It also investigated the relationship between quarantine related stressors and nightmares, sleep disturbances, and mood disorders. During our study period (i.e., by the end of May), the rate of new COVID-19 cases reached a high rate with 62,508 new cases per month, which was around 73.31% of the total number of reported cases since the beginning of the pandemic. In addition, the number of deaths during May 2020 reached 341 deaths which was equal to 67.79% of all deaths since the start of the pandemic. During the study period, the government of Saudi Arabia applied extensive measures to prevent the pandemic from spreading, especially with the past limited knowledge about the novel corona virus and the fact that there was no available approved vaccine at that time. Some of these extensive measures were complete quarantine; hence students were attending all their lessons online as well as people were prevented from visiting their relatives and families, traveling from one city to another, socially gathering in open spaces (such as parks) or closed spaces (such as malls), or attending prayers at mosques. In such escalated stress resulting from the situations at that time, our study reported that female participants had a slightly higher rate of nightmares than male participants and more frequent nightmares during the quarantine. In the literature, the higher rate of nightmares among females compared with that of males was reported to be associated with difficult childhood experiences [22,23].

Concerning the content of the nightmares, a limited number of studies were found in the literature investigating the content of nightmares. Schredl and Bulkeley, in a sample from the United States of America, showed that more than one-third of the studied sample reported threatening content in nightmares, such as being severely ill, losing loved ones to COVID-19, or nightmares not related to COVID-19 such as waking-life worries of being restricted in an extreme form [9]. An additional third of the studied sample had dreams about social distancing worries, especially the threat presented by people who fail to observe safe social distancing practices [9]. Musse et al., in a study in Brazil, reported that approximately one-third of the study sample had nightmares with pandemic-related content [10]. In addition, the contents of nightmares documented in the literature were different from the typical content reported by the majority of our participants (supernatural creatures, such as devils, or unexplained emotions, such as sudden extensive fear or guilt), which might reflect their cultural and religious concerns for not being able to perform all religious acts such as visiting mosques freely, owing to the quarantine restrictions.

Moreover, the stress of the examination period was reflected in the content of the nightmares reported by our participants, as some participants reported nightmares associated with examinations and related challenges, as well as nightmares about their future success or failure and events related to their previous school (i.e., being in familiar places or with certain friends). However, in our study, only females reported a reflection of academic pressure in their nightmares, unlike findings reported in the literature, which described a similar reflection in the dreams of both males and females [24]. Moreover, in the current study, female participants reported a higher GPA and a lower rate of failure for a given subject. Therefore, their commitments to maintaining their academic achievement might have precipitated concerns regarding their performance in the examinations, especially considering that academic attendance and examinations were conducted virtually during the pandemic, which was a new experience for most and could be a factor resulting in stressors that might then be reflected in their dreams.

Interestingly, the content of COVID-19-related nightmares was limited among the participants, contrary to our expectations. Nevertheless, approximately one-third of the participants with current nightmares reported an increased frequency of nightmares or new-onset of nightmares during the quarantine period. A recent small study reported a new and emerging complaint of nightmares among patients with post-traumatic stress disorder (PTSD) who described the COVID-19 period as “stressful but not traumatic” [25]. The 20 PTSD-affected participants reported nightmares, including content that was not associated with COVID-19 but rather with previous experiences. The authors suggested that the transient stressful experience of the pandemic might result in the re-emergence of trauma experienced in the past [25]. We hypothesized that nightmares and other stressors, such as sleep disturbances and quarantine-related worries, are risk factors for anxiety and depression. However, we believe this relationship is more likely to be a bidirectional one and that there might be a continuous loop of risk and outcome, especially if this vicious cycle is not broken by proper medical consultation and support.

Among the participants, females reported a higher frequency of becoming upset due to nightmares than males, a finding that might be associated with the reported results of females having more frequent nightmares, more difficulty falling asleep after nightmares, and a higher risk of anxiety than males. Understanding the differences in coping mechanisms with nightmares between males and females is important in planning the best medical approach to deal with nightmares, as approximately one-quarter of the female participants with nightmares sought medical advice and nearly one-third preferred the support of family or friends. On the contrary, male participants generally preferred seeking medical care, with only a minority seeking family support.

Concerning sleep disturbances, participants with nightmares experienced more frequent difficulties falling asleep. However, it was not clear whether this difficulty was the result of sleep disorders, spending time worrying and concerned being about the pandemic, or other distractors (i.e., using social media and electronic devices before bed), especially considering that social media was the preferred method of communication with friends and family, as well as a method of studying during the quarantine period [3].

Regarding the relationship between male and female groups and sleep problems, female participants with nightmares tended to have more disturbed sleep, which might be associated with their reported complaints of staying awake for some time after having a nightmare, as well as their difficulties falling asleep again. Interestingly, males with nightmares reported more frequent daytime sleepiness compared with those without nightmares. This sleepiness might result from their reported delayed sleep or early morning awakening from sleep rather than sleep disturbances during the night. Although early waking, delayed bedtime, and night sleep disturbances might shorten sleep duration, there were no reported differences in the average quantity of sleep between participants with and without nightmares.

Concerning depression and anxiety, approximately one-quarter of the participants scored positive for possible anxiety, and one-third scored positive for possible depression. In 2015, a similar rate was reported in female high school students (age 15–19 years) in the same city, Riyadh [26]. This might indicate that the COVID-19 pandemic might not trigger this level of anxiety and depression, although the severity of anxiety symptoms might have been provoked by the recent pandemic concerns, especially not being able to see their loved ones because of restrictions. Interestingly, these concerns were significantly more common in males than in females. This could perhaps be enhanced by the protective role and responsibility expected of males toward their families by our community. In addition, in our community, male university students might gather outdoors more often with their friends than female students, who tend to socialize with their friends indoors and via social media [27]. These common social worries among male participants might explain why they experienced more nightmares concerning the “loss of loved ones” than females.

The new onset of nightmares during the COVID-19 pandemic did not increase the risk of either anxiety or depression. In this context, it might be possible that the current rate of nightmares at the time of the pandemic reflected preexisting anxiety that was provoked by the stressful situations of the pandemic and quarantine. Therefore, the pandemic-related stressors might be reflected in nightmare content and an increased frequency of preexisting nightmares rather than developing new-onset nightmares. In this regard, future studies could implement screening for psychological disorders before enrolling participants and conducting follow-up with the participants to detect new-onset anxiety symptoms and nightmares.

Considering sleep disturbances and their association with mental disorders, participants with difficulty falling asleep (i.e., delayed sleep) and daytime sleepiness tended to have a higher risk of anxiety and depression. The majority of participants with nightmares, delayed bedtime, or daytime sleepiness were females. Interestingly, among our participants, the female sex independently increased the risk of anxiety but not that of depression. However, after adjusting for sleep using the global GSDS score and the presence of nightmares, the regression models for the sex variable did not remarkably affect the estimated risk of anxiety, which might possibly indicate that these two indicators of poor sleep (daytime sleepiness and difficulty falling asleep) were independent risk factors for anxiety regardless of sex type or other sociodemographic features. These two indicators might help identify students who might need more attention and psychological support during difficult times or stressful academic scenarios.

Although much research was previously conducted to evaluate mental disorders and sleep among medical students compared with other specialties, as medical students are expected to be under a lot of academic pressure [28,29,30], health and medical students in our study showed a lower rate of mental disorders, nightmares, and sleep disturbances during quarantine compared with students in other programs. This could be attributed to them being more aware of the terminology used in social media and the excellent measures our country has taken towards preventing the spread of this pandemic compared to those in non-medical or non-health-related specialties. This knowledge might have succeeded in minimizing their concerns and worries.

Although this study was a comprehensive study that covered several aspects of quarantine-related pressures (academic, social, and religious) and nightmare indicators, sleep disturbances, and the possibility of mental disorders, this study had a few limitations that need to be addressed. The use of a subjective self-completed questionnaire might have created a measurement error in the study variables. However, objective measurement of sleep, as well as a clinical diagnosis of mental disorders, would not have been feasible during the pandemic and its related quarantine and would not have been practical when our aim was to recruit a relatively large number of participants. In addition, due to the complete quarantine at the time of the study, it was not feasible to interview the participants or assess them individually to evaluate their mental health level. However, the questionnaires used in this study were previously validated questionnaires and are commonly used in clinical research. In addition, the absence of baseline descriptions for the content and frequency amongst university students in Saudi Arabia made it difficult to understand the effect of the pandemic and its related quarantine measures on nightmares accurately. In this regard, future research might be carried out to understand the effect of relieving the stress of the quarantine and the return of normal life activities on the student’s mental health.

Moreover, the study’s cross-sectional design made it difficult to identify the temporal sequence of events, especially when there was a bidirectional relationship between the study variables, such as between mental disorders, sleep problems, and nightmares. The loss of temporal sequences made it difficult to establish causality and control for confounders or to identify mediators, which might have created estimation biases.

## 5. Conclusions

In summary, the COVID-19 pandemic and its related quarantine restrictions occurred during challenging times in Saudi Arabia (i.e., during the examination period for students). Hence, the quarantine created various types of pressures among university students (i.e., academic, religious, and social pressures). Academic concerns affected the dreams of the female participants more than those of the male participants and presented in nightmare content and anxiety symptoms that were more common among the students in the sciences track. Finally, nightmares, daytime sleepiness, and delayed sleep might be helpful indicators in identifying students who might be at risk of mental disorders in various stressful situations.

## Figures and Tables

**Figure 1 ijerph-19-01776-f001:**
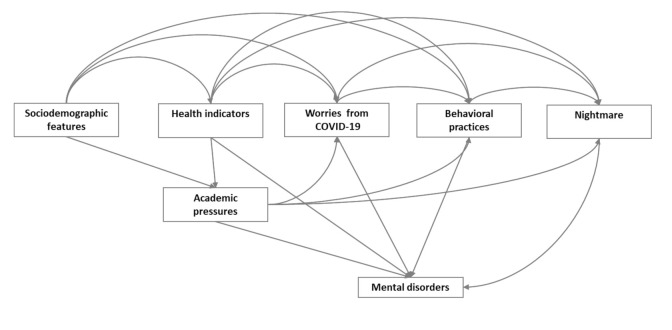
Directed acyclic graph used in building the regression model adjustment and testing the study hypothesis. The squares represent the study’s variables, and the arrows represent the direction of the relationship between the study’s variables.

**Table 1 ijerph-19-01776-t001:** The distribution of sociodemographic features of the study participants in relation to gender.

	Male		Female		Total	X^2^	Df	*p*
	*n* = 132	%	*n* = 236	%	*n* = 368			
Family income								0.857 *
Low	0	0.00	2	84.7	2			
Middle low	52	39.39	93	39.40	145			
Middle	3	2.27	7	2.96	10			
Middle high	60	45.45	110	46.61	170			
High	17	12.87	24	10.16	41			
Marital status								0.028 *
Single	124	93.93	232	98.30	356			
Married	8	6.06	44	18.64	12			
School						50.90	3	<0.001
Science and architect	60	45.45	177	75	237			
Medicine and Health	5	3.78	13	5.50	18			
Humanity and art	57	43.18	26	11.01	83			
Administration and management	10	7.57	20	8.47	30			
Father education						10.17	4	0.038
Primary education	8	6.06	10	4.23	18			
Secondary education	20	15.15	16	6.77	36			
High school	31	23.48	72	30.50	103			
University	45	34.09	98	41.52	143			
Postgraduate	28	21.21	40	16.94	68			
Mother education						0.58	4	0.965
Primary education	14	10.60	22	9.32	36			
Secondary education	15	11.36	27	11.44	42			
High school	39	29.54	78	33.05	117			
University	56	42.42	96	40.67	152			
Postgraduate	8	6.81	13	5.50	21			
Academic year						35.93	5	<0.001
1st	30	22.72	25	45.45	55			
2nd	32	56.14	25	43.86	57			
3rd	26	37.68	43	62.32	69			
4th	17	16.50	86	83.50	103			
5th	16	32.65	33	67.35	49			
6th	11	31.43	24	68.57	35			
Academic failure						20.22	1	<0.001
No	92	30.55	209	69.44	301			
Yes	40	59.70	27	40.30	67			
GPA						46.16	3	<0.001
D	7	77.78	2	22.22	9			
C	35	76.09	11	23.91	46			
B	49	29.34	118	70.66	167			
A	41	28.08	105	71.92	146			
Chronic disease						0.679	1	0.4101
No	122	36.53	212	63.47	334			
Yes	10	29.41	24	70.59	34			
Smoking						52.36	1	<0.001
No	88	28.21	224	71.79	312			
Yes	44	78.57	12	21.43	56			
Regular physical exercise						0.142	1	0.707
No	92	36.51	160	63.49	252			
Yes	40	34.48	76	65.52	116			

******p*-value comes from Fisher exact test.

**Table 2 ijerph-19-01776-t002:** Distribution of nightmare indicators amongst study participants concerning gender (*n* = 368).

	Male		Female		Total	X^2^	Df	*p*
	*n* = 132	%	*n* = 236	%	*n* = 368			
Presence of nightmares						6.55	1	0.010
No	101	40.24	150	59.76	251			
Yes	31	26.50	86	73.50	117			
New onset of nightmare or increased frequency in relation to COVID-19						7.29	1	0.007
No	122	38.61	194	61.39	316			
Yes	10	19.23	42	80.77	52			
Change in the frequency of nightmares before and after COVID-19						14.35	2	0.001
Did not change	102	42.68	137	57.32	239			
Increased or new-onset	10	19.23	42	80.77	52			
Decrease	20	25.97	57	74.03	77			
Frequency of nightmares during COVID-19						19.23	3	<0.001
No nightmare	91	45.73	108	54.27	199			
Daily	5	35.71	9	64.29	14			
>=3 times per week	8	21.62	29	78.38	37			
<3 times per week	28	23.73	90	76.27	118			
Nightmare theme during COVID-19								<0.001 *
No nightmares	91	45.73	108	54.27	199			
Related to the pandemic	7	5.30	15	6.36	22			
◦ Catching COVID-19 disease	2	25.00	6	75.00	8			
◦ Loss of loved ones	4	57.14	3	42.86	7			
◦ Death of themselves	1	20.00	4	80.00	5			
◦ Diseases in general	0	0.00	2	2	2			
◦ Others not related to COVID-19	34	23.13	113	76.87	147			
◦ Falling or driving very fast	2	28.57	5	71.43	7			
◦ Fighting	0	0.00	2	100	2			
◦ Lost or kidnapped	0	0.00	4	100	4			
◦ Academic life and exams	0	0.00	5	100	5			
◦ Natural disaster	2	66.67	1	33.33	3			
◦ Future or past events	0	0.00	3	100	3			
◦ Supranatural creatures or unexplained emotions	18	26.47	50	73.53	68			
◦ Cannot remember	1	11.11	8	88.89	9			
Bothered by nightmares								<0.001 *
No nightmare s	91	45.73	108	54.27	199			
Bother me a lot	4	18.18	18	81.82	22			
Bother me a little	8	22.22	28	77.78	36			
Does not bother me	19	25.00	57	75.00	76			
Does not bother me at all	10	28.57	25	71.43	35			
Back to sleep after a nightmare						20.95	3	<0.001
No nightmares	91	45.73	108	54.27	199			
Difficulty going back to sleep	5	13.16	33	86.84	38			
Stay awake some time before going back to sleep	22	27.85	57	72.15	79			
Sleep immediately	14	26.92	38	73.08	52			
Overcome nightmares								<0.001 *
No nightmares	91	45.73	108	54.27	199			
Talk to family members	2	5.00	38	95.00	40			
Talk to friends	11	26.83	30	73.17	41			
Consult a physician	25	32.47	52	67.53	77			
Internet search	0	0.00	1	100	1			
No action	3	30.00	7	70.00	10			

* *p*-value resulted from Fisher exact test, (x^2^) = chi-square test, (DF) = degree of freedom, (*p*) = *p*-value.

**Table 3 ijerph-19-01776-t003:** Distribution of sleep disturbance components measured using GSDS concerning gender and presence of nightmares (*n* = 368).

	Male (*n* = 132)				Female (*n* = 236)				Total (*n* = 368)			
	No Nightmares		With Nightmares		No Nightmares		With Nightmares		No Nightmares		With Nightmares	
	*n* = 91	%	*n* = 41	%	*n* = 108	%	*n* = 128	%	*n* = 199	%	*n* = 169	%
Difficulty falling asleep	X^2^(1) = 1.18, *p* = 0.290				X^2^(1) = 21.95, *p* < 0.001				X^2^(1) = 21.85, *p* < 0.001			
<=times/week	49	73.13	18	26.87	63	63.64	36	36.36	112	67.47	54	32.53
>3 times/week	42	64.62	23	35.38	45	32.85	92	67.15	87	43.07	115	56.93
Waking up during sleep	X^2^(1) = 0.57, *p* = 0.452				X^2^(1) = 6.589, *p* = 0.010				X^2^(1) = 9.17, *p* = 0.002			
<=times/week	53	71.62	21	28.38	55	55.56	44	44.44	108	62.43	65	37.57
>3 times/week	38	65.52	20	34.48	53	38.69	84	61.31	91	46.67	104	53.33
Waking up too early from sleep	X^2^(1) = 7.64, *p* = 0.006				X^2^(1) = 2.60, *p* = 0.107				X^2^(1) = 12.55, *p* < 0.001			
<=times/week	61	78.21	17	21.79	50	52.08	46	47.92	111	63.79	63	36.21
>3 times/week	30	55.56	24	44.44	58	41.43	82	58.57	88	45.36	106	54.64
Poor Quality of sleep	X^2^(1) = 0.03, *p* = 0.855				X^2^(1) = 0.69, 0.406				X^2^(1) = 0.66, *p* = 0.416			
<=times/week	34	68.00	16	32.00	39	42.39	53	57.61	73	51.41	69	48.59
>3 times/week	57	69.51	25	30.49	69	47.92	75	52.08	126	55.75	100	44.25
Abnormal quantity of sleep	X^2^(1) = 2.55, *p* = 0.110				X^2^(1) = 1.37, *p* = 0.242				X^2^(1) = 3.46, *p* = 0.063			
<=times/week	40	43.96	12	29.27	46	42.59	45	35.16	86	60.14	57	39.86
>3 times/week	51	56.04	29	70.73	62	57.41	83	64.84	113	50.22	112	49.78
Use of substances to help induce sleep	*p* = 0.173 *				*p* = 0.542 *				*p* = 0.269 *			
<=times/week	89	97.80	38	92.68	108	100.00	127	99.22	197	54.42	165	45.58
>3 times/week	2	2.20	3	7.32	0	0.00	1	100	2	33.33	4	66.67
Day-time sleepiness	X^2^(1) = 5.7115, *p* = 0.017				X^2^(1) = 5.64, *p* = 0.018				X^2^(1) = 14.51, *p* < 0.001			
<=times/week	66	72.53	21	51.22	64	59.26	56	43.75	130	62.80	77	37.20
>3 times/week (161)	25	27.47	20	48.78	44	40.74	72	56.25	69	42.86	92	57.14

* *p*-value resulted from Fisher exact test, (x^2^) = chi-square test, (*p*) = *p*-value.

**Table 4 ijerph-19-01776-t004:** Results of several adjusted and unadujsted logistic regression models that estimated the risk of having anxiety and depression in relation to several nightmare indicators.

		Number of Patients with Anxiety (*n* = 100)	Anxiety		Number of Patients with Depression (*n* = 127)	Depression	
			Unadjusted OR (CI)	Adjusted * OR (CI)		Unadjusted OR (CI)	Adjusted * OR (CI)
Presence of nightmares	No	44	Ref.		71		
	Yes	56	**4.32** (**2.65, 7.03**)	**4.13** (**2.49, 6.85**)	56	**2.33** (**1.48, 3.67**)	**2.15** (**1.34, 3.45**)
New onset of nightmares related to COVID-19	No	85	Ref.		110	Ref.	
	Yes	18	1.10 (0.58, 2.11)	1.05 (0.54, 2.07)	17	0.91 (0.49, 1.70)	0.94 (0.49, 1.81)
Change in the frequency of nightmares before and after COVID-19	Did not change	29	Ref.		54		
	Increased	15	1.57 (0.79, 3.09)	1.47 (0.73, 2.97)	17	1.20(0.63, 2.28)	**3.22** (**1.66, 6.25**)
	Decreased	36	**3.40** (**1.97, 5.88**)	**3.02** (**1.72, 5.29**)	41	**2.81** (**1.66, 4.76**)	1.58 (0.95, 2.62)
Frequency of nightmares during COVID-19	No nightmare	29	Ref.		54		
	>=3 times per week	28	**7.14** (**3.62, 14.05**)	**6.89** (**3.41, 13.89**)	28	**3.27** (**1.73, 6.16**)	**3.22** (**1.66, 6.25**)
	<3 times per week	43	**3.36** (**1.95, 5.79**)	**3.23** (**1.84, 5.69**)	45	1.66 (1.02, 2.69)	1.58 (0.95, 2.62)
Nightmare theme during COVID-19	No nightmare	29	Ref.		54		
	COVID-19 related	12	**7.03** (**2.7, 17.78**)	**3.77** (**2.21, 6.44**)	10	**2.01** (**1.28, 3.16**)	**1.92** (**1.19, 3.09**)
	Not related to COVID-19	59	**3.93** (**2.35, 6.57**)	**6.84** (**2.62, 17.83**)	63	**2.24** (**1.28, 3.16**)	2.28 (0.90, 5.79)
Bothered by nightmares	No nightmare	29	Ref.		54		
	Does not bother me	34	**2.59** (**1.47, 4.55**)	**2.56** (**1.43, 4.59**)	35	1.24 (0.74, 2.05)	1.23 (0.73, 2.10)
	Bothers me	37	**10.33** (**5.31, 20.08**)	**9.33** (**4.71, 18.49**)	38	**5.10** (**2.73, 9.53**)	**4.57** (**2.39, 8.74**)
Back to sleep after a nightmare	No nightmare	29	Ref.		54		
	Difficulty going back to sleep	28	**16.41** (**7.21, 37.36**)	**14.49** (**6.19, 33.92**)	29	**8.65** (**3.45, 19.46**)	**7.79** (**3.35, 18.10**)
	Stay awake some time before going back to sleep	25	**2.71** (**1.47, 5.02**)	**2.83** (**1.50, 5.34**)	26	**1.32** (**3.85, 19.46**)	1.39 (0.78, 2.49)
	Sleep immediately	18	**3.10** (**1.55, 6.21**)	**2.94** (**1.44, 5.99**)	18	1.42 (0.74, 2.73)	1.32 (0.67, 2.59)
Overcome nightmares	No nightmare	29	Ref.		54		
	Talk to family members	16	**3.91** (**1.85, 8.23**)	**3.66** (**1.64, 8.19**)	17	1.98 (0.99, 3.99)	1.8 (0.86, 1.92)
	Talk to friends	23	**7.49** (**3.60, 15.57**)	**7.71** (**3.61, 16.44**)	23	**3.43** (**1.72, 6.85**)	**3.37** (**1.65, 6.92**)
	Consult a physician	30	**3.66** (**2.01, 6.69**)	**3.44** (**1.85, 6.39**)	29	1.59 (0.91, 2.77)	1.54 (0.87, 2.75)
	No action	2	1.47 (0.29, 7.25)	1.47 (0.29, 7.47)	4	1.79 (0.49, 6.59)	1.71 (0.45, 6.57)

* Adjusted for academic year, gender, socioeconomic status, chronic diseases, smoking, exercise. Bold fonts indicate significant results (*p*-value < 0.0).

**Table 5 ijerph-19-01776-t005:** Results of logistic regression models that estimated the risk of having anxiety and depression in relation to sleep disturbance components.

		Number of Patients with Anxiety (*n* = 100)	Anxiety		Number of Patients with Depression (*n* = 127)	Depression	
			Unadjusted OR (CI)	Adjusted * OR (CI)		Unadjusted OR (CI)	Adjusted * OR (CI)
Difficulty falling asleep	<=times/week	25	Ref.		46		
	>3 times/week	75	**3.33 (1.99, 5.56)**	**3.55 (2.09, 6.06)**	81	**1.75 (1.12, 2.71)**	**1.76 (1.11, 2.78)**
Waking up during sleep	<=times/week	30	Ref.		41		
	>3 times/week	70	**2.67 (1.63, 4.36)**	**2.46 (1.49, 4.07)**	86	**2.54 (1.62, 3.98)**	**2.35 (1.48, 3.74)**
Waking up too early from sleep	<=times/week	32	Ref.		51		
	>3 times/week	68	**2.39 (1.48, 3.89)**	**2.24 (1.36, 3.69)**	76	1.55 (0.047, 1.00)	1.47 (0.93, 2.31)
Poor Quality of sleep	<=times/week	45	Ref.		53		
	>3 times/week	55	0.69 (0.43, 1.11)	0.64 (0.39, 1.05)	74	0.82 (0.53, 1.27)	0.79 (0.50, 1.26)
Abnormal quantity of sleep	<=times/week	25	Ref.		31		
	>3 times/week	75	**2.36 (1.41, 3.94)**	**2.52 (1.48, 4.27)**	96	**2.69 (1.67, 4.33)**	**2.99 (1.81, 4.92)**
Use of substances to help induce sleep	<=times/week	99	Ref.		125		
	>3 times/week	1	0.53 (0.061, 4.60)	0.48 (0.05, 4.44)	2	0.95 (0.17, 5.25)	0.88 (0.15, 5.19)
Daytime sleepiness	<=times/week	22	Ref.		44		
	>3 times/week	78	**7.90 (4.61, 13.55)**	**7.48 (4.32, 12.96)**	83	**3.94 (2.50, 6.21)**	**3.73 (2.34, 5.96)**

* Adjusted for academic year, gender, socioeconomic status, chronic diseases, smoking, exercise. Bold fonts indicate significant results (*p*-value < 0.05).

## Data Availability

Data available on request due to ethical restrictions. The data presented in this study are available on request from the corresponding author.

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
