# Peer review of "Nightmares’ Pattern and Predictors among a Saudi University Students during COVID-19 Pandemic"

_ijerph, 2022, doi:10.3390/ijerph19031776_

Round 1

Reviewer 1 Report

Dr Alghandi et al must be acknowledged for conducting a comprehensive study covering several aspects of quarantine-related process (academic, social, religious) and designing  self-completed questionnaires with items regarding socio-demographic features and health indications, nightmares, sleep disturbances and possible anxiety and depressive mood.

The questionnaires, GSDS, Generalized Anxity Disorder 2 scale and Patient Health Questionnaire are relevant                                                                 The statistics are sound and informative. The results are numerous and of interest. The discussion is well conducted despite a certain number of speculations to explain some results

However, some criticisms apply

1) We learn that the study was performed during 2 weeks, within the quarantine period, that is between 23 March 2020 and 28 May 2020. Please mention  the exact period of the survey, beginning, middle or end of the quarantine

2) It  is unfortunate that the authors do not inform the reader                         - about   the severity of the pandemia at the time of the study, number of COVID-19 new cases per 100.000 inhabitants in Ryad, number of hospitalized subjects, number of patients in the resuscitation unit                                       - about  the type of the variant                                                                          - about  the degree of quarantine undergone by the students: students having their academic examination period, but not allowed to visit their family, their friends and the mosque. Please be more precise about the restrictions: meals, dormitories,  allowance to walk in the street, librairies?       All these details are important to judge the severity of the quarantine and to compare the current study with the studies run in other countries   

Details                                                                                                              By definition it was not possible to submit these students to a psychological examination or test prior to the pandemia. However, students could have been interviewed on their personal mental equilibrium at the time of the survey, even afterwards

Introduction: "Saoudi Arabia's efforts to control the spread of the coronavirus disease 2019 are attracting global attention after researchers across the country demonstrated the success of a chain of adapted measures. The World Health Organization has commanded the Saoudi exemple as a "success story" during the G20 leaders summit held on November 2020"   Please avoid this kind of  self-satisfaction as regards  your own country

Author Response

Please find the attachments of the reviewers comments below.

Reviewer 2 Report

  1. The manuscript is too lengthy. There is no need to describe all associations between each item in the questionnaire. Keep only important findings.
  2. Lack of interpretation of the acyclic graph.
  3. Line 123: Nightmare was not identified by a physician. Prevent the use of "diagnosis."
  4. Line 110-114: revise the power statement.
  5. There have been too many papers about the COVID-19 and nightmare. What's new?
  6.  Are the contents of the nightmare during the pandemic important and will be different from the nightmare without the pandemic? 

Author Response

(The authors gave the same response as above.)

Reviewer 3 Report

Thank you for the opportunity to review a very interesting topic. Congratulations to the authors of your interest in the topic. In the discussion, I suggest citing more research, e.g. Ozga D. Search for factors of anxiety incidence in midwifery students during the COVID-19 pandemic, to show the problem globally. 

Author Response

(The authors gave the same response as above.)

Round 2

Reviewer 2 Report

The authors have answered my questions.